# Long-Term Incidence of Total Knee Arthroplasty after Open Reduction and Internal Fixation of Proximal Tibial and Distal Femoral Fractures: A Nationwide Cohort Study

**DOI:** 10.3390/jcm10235685

**Published:** 2021-12-02

**Authors:** Kuang-Ting Yeh, Wen-Tien Wu, Ru-Ping Lee, Chen-Chie Wang, Jen-Hung Wang, Ing-Ho Chen

**Affiliations:** 1Department of Orthopedics, Hualien Tzu Chi Hospital, Buddhist Tzu Chi Medical Foundation, Hualien 970473, Taiwan; micrograft@tzuchi.com.tw (K.-T.Y.); timwu@tzuchi.com.tw (W.-T.W.); 2School of Medicine, Tzu Chi University, Hualien 970374, Taiwan; xaiver-wang@yahoo.com.tw; 3Institute of Medical Science, Tzu Chi University, Hualien 970374, Taiwan; fish@gms.tcu.edu.tw; 4Department of Orthopedic Surgery, Taipei Tzu Chi Hospital, Buddhist Tzu Chi Medical Foundation, New Taipei City 231405, Taiwan; 5Department of Medical Research, Hualien Tzu Chi Hospital, Buddhist Tzu Chi Medical Foundation, Hualien 970473, Taiwan; paulwang@tzuchi.com.tw

**Keywords:** total knee arthroplasty, proximal tibial fracture, distal femoral fracture, nationwide-based cohort study, propensity score matching method

## Abstract

Knee fractures often require open reduction internal fixation (ORIF) for knee function recovery. More than one fifth of patients with knee fractures subsequently develop posttraumatic arthritis, and over 5% of them need total knee arthroplasty (TKA). We conducted this nationwide retrospective cohort study using the data of 2,000,000 people in the general population who received TKA and were followed up in the 17-year period 2001–2017, through random sampling of the Taiwan National Health Insurance Research Database. We matched the ORIF and non-fracture groups by a propensity score, based on age, sex, index date of surgery, and comorbidities enrolled in CCI calculated at a 1:1 ratio. The average proportion of subjects receiving TKA after ORIF for distal femoral or proximal tibial fractures was 2.0 per 1000 person-years, which was significantly higher than that in the non-fracture group. Patients aged 20–65 years and males represented a significantly higher proportion of subjects receiving TKA after ORIF than that in the non-fracture group. The proportion of subjects receiving TKA for the 20–65-year subgroup of the ORIF group was 4%, and that for the male subgroup was 1.5%; both rates increased over the 17-year follow-up period. More aggressive intraoperative and postoperative adjuvant therapies may be necessary for these subgroups.

## 1. Introduction

Fractures of the distal femur, proximal tibia, and patella often engender inferior knee function [1], poor health-related quality of life [2], and further development of posttraumatic knee arthritis [3]. Posttraumatic knee arthritis occurs in approximately 23% to 36% of cases following intra-articular fractures [4]. This incidence does not appear to have decreased over time, despite advancements in fracture management and care strategies [5]. Moreover, posttraumatic knee arthritis is a critical risk factor for an increased rate of total knee arthroplasty (TKA), in addition to a relatively long human lifespan and an increased rate of obesity [6]. A systematic review suggested the optimal age for TKA was in the early 70s because patients within this age range could achieve the optimal passive range of motion after surgery [7]. Moreover, patients younger than 55 years were reported to have a significant effect on major aseptic revision rates [8]. A study also revealed that the rates of in-hospital adverse events after TKA decreased gradually from 2010 to 2017; however, critical risk factors for severe adverse events included the male sex [9]. Another study reported that the recurrence rate of postoperative flexion contracture was significantly higher in men than in women [10]. Therefore, age and sex differences, which are significantly correlated with future knee function and quality of life, appear to be critical factors for TKA-related knee function. Nevertheless, few large population cohort studies have focused on the influence of these factors on TKA rates after knee trauma. The use of conventional clinical approaches to investigate the relationship between open reduction internal fixation (ORIF) surgery for proximal tibial or distal femoral fractures and subsequent TKA is challenging because TKA is an infrequent complication after ORIF surgery. To fill the aforementioned research gap, we conducted this population-based retrospective study to evaluate the proportion of subjects receiving TKA and the age- and sex-related risk factors for TKA following ORIF surgery, for distal femoral or proximal tibial fractures.

## 2. Materials and Methods

Our study protocol was approved by the Research Ethics Committee of Hualien Tzu Chi Hospital (IRB108-242-C). We collected data from the Taiwan National Health Insurance Research Database (NHIRD), which is provided by the National Health Insurance Administration, Ministry of Health and Welfare, and managed by the National Health Research Institutes (NHRI). The NHIRD holds data on 99.7% of the population (nearly 23 million people) in Taiwan. Longitudinal Health Insurance Databases (LHID2000), which randomly sampled 2 million beneficiaries from the original NHIRD for the year 2000, was adopted for this study. The representativeness of LHIDs has been validated by NHRI. We included the data of patients who underwent ORIF surgery for distal femoral or proximal tibial fractures and subsequently received TKA between 1 January 2001 and 31 December 2017 (ORIF group). For the non-fracture group, we also collected the data for 2,000,124 people in the general population through random sampling from the NHIRD (Figure 1). Relevant disease and surgical codes are listed in Appendix A. The comorbidities included in this study were based on those used in the Charlson Comorbidity Index (CCI) [11]. The case definition of each comorbidity was based on diagnostic codes, which have been employed and validated in other studies using claims databases. These comorbidities were identified by the presence of either at least two entries of diagnostic codes in the outpatient records, or one occurrence of discharge codes in hospitalization records, in the year before the index date.

We used the following methodology: A propensity score matching method was applied to reduce selection bias between the study groups. Age, sex, index year date of surgery, and comorbidities enrolled in CCI calculated at a 1:1 ratio were selected as independent variables. The Greedy method was used for matching, at a 1:1 ratio, the study groups, with a caliper width of 0.2-fold the standard deviation of the propensity score between the study groups. The study outcome was subsequent to TKA performed after index surgery for distal femoral or proximal tibial factures. All available data were included in this study, and no additional unpublished data were included. The subgroup analysis was then performed by dividing all the patients into two groups by age (20–65 years and >65 years) and by sex (male and female). Interaction tests were employed to determine the subgroup effects of age and sex on the TKA risk. 

Continuous variables are presented as means and standard deviations, and categorical variables are presented as number of cases and percentages. Continuous between-group variables were compared using Student’s *t*-test, and categorical variables were assessed with a chi-square test. These tests were used to compare the characteristics of both groups. Data were evaluated using univariable and multivariable Cox regression analyses. Survival curves were estimated according to the Kaplan–Meier procedure, and groups were compared with use of the log-rank test. All statistical analyses were performed using SAS version 9.4 (SAS Institute Inc., Cary, NC, USA) and Stata 16.1 (StataCorp, College Station, TX, USA). We considered *p*-values of <0.05 as statistically significant.

## 3. Results

The 1:1 propensity score matching yielded 32,592 patients in the ORIF group and 32,592 individuals in the non-fracture group (Table 1). The TKA rates in the ORIF and non-fracture groups were 2.0 and 1.6 per 1000 person-years, respectively (Table 2). From the Cox proportional hazard regression model, with adjustments for all baseline characteristics shown in Table 1, we observed that the adjusted hazard ratio (aHR) derived for the ORIF group relative to the non-fracture group was 1.23 (1.07–1.41; Table 2), which was determined to be significant (*p* = 0.004; Table 2). The further subgroup analysis revealed that patients aged 20–65 years in the ORIF group had a significantly higher risk of TKA than their counterparts in the non-fracture group (aHR = 1.94, 1.51–2.48). Male patients in the ORIF group also had a significantly higher risk of TKA than their counterparts in the non-fracture group (aHR = 1.56, 1.19–2.04; Table 3). The effects of age and sex on the TKA risk were significant based on *p* for interaction results (Table 3). The proportion of subjects receiving TKA in the 20–65-year subgroups of the ORIF and non-fracture groups increased linearly over time. A significantly higher risk of TKA was also observed in the ORIF group when compared with the non-fracture group (Figure 2A). The same distribution was also noted in the male subgroups of both the non-fracture and ORIF groups (Figure 2B). After 17 years, the proportion of subjects receiving TKA for the 20–65-year subgroup of the ORIF group was 4%, and that for the male subgroup was 1.5% (Figure 2).

## 4. Discussion

Distal femoral and proximal tibial fractures often include intra-articular injury and further lead to posttraumatic osteoarthritis. Most cases of degenerative arthritis of the knee in patients younger than 50 years are due to posttraumatic osteoarthritis [12]. Reports of the management of geriatric knee fractures have increased in recent decades because people now have a longer lifespan. TKA should be performed on patients with advanced knee arthritis and unacceptable knee function. According to patient-reported outcomes from a national cross-sectional study, female patients aged older than 40 years had high odds ratios for poor functional outcomes in long-term follow-up after a knee fracture [13]. A previous study reported that a major factor for poor functional status after trauma was posttraumatic osteoarthritis, the incidence of which was as high as 75% [14]. Moreover, a 2014 Canadian nationwide matched population-based cohort study reported that the overall incidence of tibial plateau fractures leading to end-stage posttraumatic osteoarthritis that required further TKA was 7.3% at 10 years after ORIF; relevant risk factors included increasing age, bicondylar fracture, concomitant meniscal tear injury, increased medical comorbidities, and the female sex [15]. A 20-year Danish nationwide cohort study revealed that patients with knee fractures had a 1.6 times greater risk of TKA at 3 years after trauma and throughout their lifetimes compared to other individuals; the study indicated that surgical treatment of knee fractures, or external fixation, remained the major relative risk factor [16]. In our study, the TKA rate after ORIF for proximal tibial and distal femoral fractures was 2.0 per 1000 person-years, and the 17-year incident rate of TKA was less than 4%, which are comparable to those reported in the literature [15,16,17]. This indicates that, although severe knee fractures that require ORIF may be correlated with a high risk of TKA, adequate ORIF for knee fractures is still a crucial procedure for preserving knee function after traumatic fractures.

The principle of ORIF for intra-articular knee fractures includes anatomical articular surface restoration, condylar width restoration, mechanical axis alignment restoration, joint stability reconstruction, and soft tissue repair [18]. Currently, strategies extensively applied for treating or managing distal femoral and proximal tibial fractures involve locked plating of fractures and minimally invasive approaches, such as anatomical reduction of fractures, which can considerably reduce the development of advanced posttraumatic knee osteoarthritis and facilitate postoperative functional recovery [19,20,21]. This thus explains the lower TKA rates in our cohort study compared with those in the literature. In our study, the TKA rates observed for male patients and younger patients were nearly two times higher than those observed for the non-fracture group. The possible reasons for this finding are multifactorial, including working status, early return to work because of economic factors, or poor compliance with physical therapy. Such patients may require more aggressive postoperative management approaches and follow-up protocols, including early detection of the progression of posttraumatic arthritis with serum biomarker levels [22], radiographic studies [23], prompt intervention with anti-inflammatory and regeneration therapy [24], aggressive rehabilitation programs for muscle strength around the knee joint, and adequate protective braces suitable for early return to work after ORIF.

We originally matched the fracture and non-fracture groups through a 1:2 ratio by age, sex, and index date, and the analytical results of this matching method are represented as Appendix A. To further eliminate significant differences in the comorbidities between the groups, we then matched the ORIF and non-fracture groups by propensity score, age, sex, index date of surgery, and comorbidities enrolled in CCI calculated at a 1:1 ratio. The results of both methods were similar, and this indicates the validity of our results.

A strength of this study is the large sample size; the comprehensive coverage of the National Health Insurance system (covering > 95% of the Taiwanese population) may have minimized selection and nonresponse biases. However, this study has some limitations. First, the data were acquired from NHIRD, which registers only surgical fee codes, and the severity or classification of the fractures could not be determined for further correlation with the rate of future TKA incidence. Second, the data on lifestyle factors, personal characteristics, and detailed postoperative adjuvant treatment, which may be influential sources of bias, were unavailable because the NHIRD does not provide this information. Third, because our results are based on data from Taiwan, the present findings may not be directly generalizable to Caucasian or African populations.

## 5. Conclusions

The average proportion of subjects receiving TKA after ORIF for distal femoral or proximal tibial fractures was 2.0 per 1000 person-years and was significantly higher than that in the non-fracture group, especially in the subgroups comprising individuals aged 20–65 years and men. More aggressive intraoperative and postoperative adjuvant therapies and protocols for these subgroups may be necessary. The proportion of subjects receiving TKA after knee fracture increased gradually during the 17-year follow-up period. The long-term proportion of subjects receiving TKA in our study is lower than that in the literature, and this difference is attributable to race differences or recent surgical advances.

## Figures and Tables

**Figure 1 jcm-10-05685-f001:**
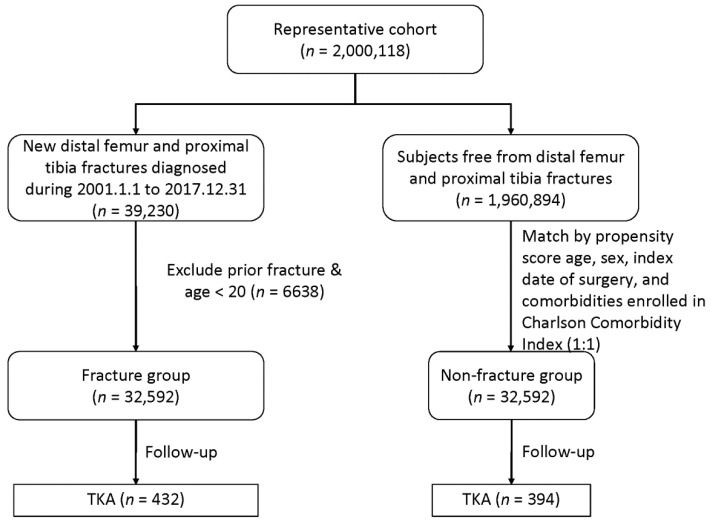
The study flow chart.

**Figure 2 jcm-10-05685-f002:**
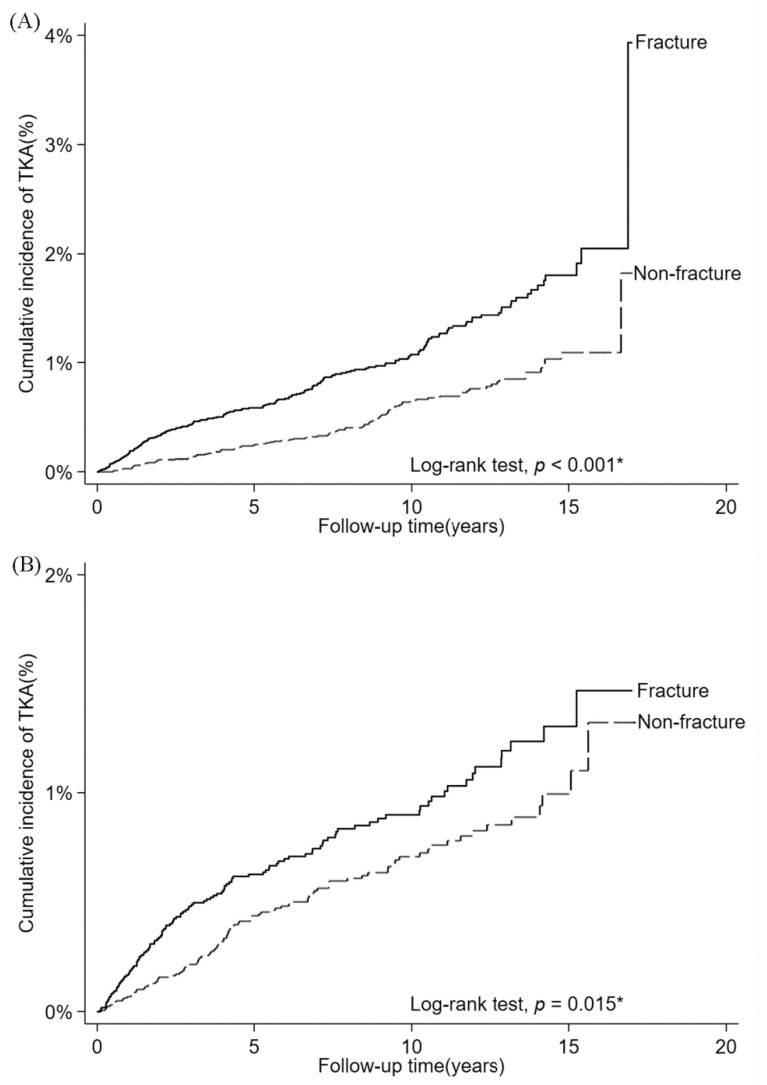
The proportion of subjects receiving TKA of the fracture group and the non-fracture group. (**A**) The 20–65-year subgroups of the ORIF and non-fracture groups. (**B**) The male subgroups of the ORIF and non-fracture groups. * A value of *p* < 0.05 was considered statistically significant after test.

**Table 1 jcm-10-05685-t001:** Baseline characteristics and comorbidity.

	Propensity Score Matching (1:1)		
	Non-Fracture (*n* = 32,592)	Fracture (*n* = 32,592)	*p*-Value
Age (y/o)	58.75 ± 20.01	58.46 ± 19.92	0.058
Age Group			0.072
<65 y/o	17,985 (55.2%)	18,213 (55.9%)	
≧65 y/o	14,607 (44.8%)	14,379 (44.1%)	
Male (%)	16,849 (51.7%)	16,873 (51.8%)	0.851
Comorbidity			
HTN	10,134 (31.1%)	10,029 (30.8%)	0.374
DM	5803 (17.8%)	5747 (17.6%)	0.566
Hyperlipidemia	3377 (10.4%)	3440 (10.6%)	0.420
Chronic renal failure	936 (2.9%)	1015 (3.1%)	0.069
CAD	2835 (8.7%)	2879 (8.8%)	0.542
CVA	3183 (9.8%)	3128 (9.6%)	0.466
Alcohol-induced mental disorders	50 (0.2%)	70 (0.2%)	0.068
Alcohol dependence syndrome	84 (0.3%)	112 (0.3%)	0.045 *
Drug dependence	37 (0.1%)	44 (0.1%)	0.436
Chronic liver disease	1636 (5.0%)	1697 (5.2%)	0.278
Iron deficiency anemia	310 (1.0%)	384 (1.2%)	0.005 *
Depression	1432 (4.4%)	1402 (4.3%)	0.564
Dementia	1233 (3.8%)	1204 (3.7%)	0.549
Peripheral vascular disease	330 (1.0%)	385 (1.2%)	0.039 *

y/o: years old. Data are presented as n and percentage. * A value of *p* < 0.05 was considered statistically significant after test.

**Table 2 jcm-10-05685-t002:** Risk of TKA in patients with and without fracture.

	Propensity Score Matching (1:1)	
	Fracture	
	Yes	No
Patient numbers	32,592	32,592
TKA cases	432	394
Person-years	219,264	244,653
Incidence rate ^a^	2.0	1.6
Univariable model		
crude HR (95% CI)	1.16 (1.01–1.33)	1 (ref.)
*p*-value	0.035 *	
Multivariable model ^b^		
aHR (95% CI)	1.23 (1.07–1.41)	1 (ref.)
*p*-value	0.004 *	

aHR: adjusted hazard ratio; CI: confidence interval; HR: hazard ratio; ref: reference. ^a^ Per 1000 person-years. ^b^ Multivariable Cox proportional hazard regression model with adjustment for all baseline characteristics shown in Table 1. * A value of *p* < 0.05 was considered statistically significant after test.

**Table 3 jcm-10-05685-t003:** Subgroup analysis of Cox’s regression model for the association between fracture and TKA.

Variables	Propensity Score Matching (1:1)				
	Crude HR ^a^ (95% CI)	*p*-Value	Adjusted HR ^a^ (95% CI)	*p*-Value	*p* for Interaction
Main model					
Non-fracture	1.00		1.00		
Fracture	1.16 (1.01–1.33)	0.035 *	1.23 (1.07–1.41)	0.004 *	
Age					<0.001 *
20–65 y/o					
Non-fracture	1.00		1.00		
Fracture	1.94 (1.52–2.49)	<0.001 *	1.94 (1.51–2.48)	<0.001 *	
≧65 y/o					
Non-fracture	1.00		1.00		
Fracture	0.97 (0.82–1.15)	0.746	0.96 (0.81–1.14)	0.634	
Gender					0.075
Male					
Non-fracture	1.00		1.00		
Fracture	1.39 (1.07–1.82)	0.015 *	1.56 (1.19–2.04)	0.001 *	
Female					
Non-fracture	1.00		1.00		
Fracture	1.08 (0.92–1.27)	0.368	1.13 (0.96–1.33)	0.131	

CI, confidence interval; HR, hazard ratio. ^a^ Cox’s proportional hazards model. * A value of *p* < 0.05 was considered statistically significant after test.

## Data Availability

Our data are third-party data, which are now managed by the Ministry of Health and Welfare. Others can access these datasets by application and following approval by our government. Our data are restricted by National Health Insurance, Department of Health of Taiwan. The government made the rule that all data be returned after this study finished. The data cannot be studied without approval. If there are further requests for the data, one should apply for the data from our National Health Insurance. The rules of how to apply for and use these data can be found at https://dep.mohw.gov.tw/DOS/mp-113.html. The authors confirm they had no special privileges or access to the data and that all interested, qualified researchers may access the data using the application process described.

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
