# Peer review of "Long-Term Incidence of Total Knee Arthroplasty after Open Reduction and Internal Fixation of Proximal Tibial and Distal Femoral Fractures: A Nationwide Cohort Study"

_jcm, 2021, doi:10.3390/jcm10235685_

Round 1
Reviewer 1 Report
This is a study of patients with ORIF and a possible later TKA. Patients with ORIF are matched with patients without ORIF from the general population in order to study differences in the cumulative incidence rate of TKA. I have a few comments:
- I do not se the need for the first attempt with exact matching based on age sex, and year of surgery. It is concluded that this sample was unbalanced considering comorbidity wherefore propensity scores were used instead. The results are similar, however, wherefore I think the manuscript would be clearer without the initial attempt.
- The propensity score procedure is poorly described. Which method was used (logistic regression, GBM etc)? Greedy matching? Caliper? And especially, what variables were included?
- Who is included in NHIRD? The whole population or just patients who received care during the studied period? If so, at what level? This is rather important for the interpretation of the study!
- What was the timing of the comorbidity listed in table 1? Were they all present simultaneously as the ORIF or is this based on some look-back period using some records linkage? Or could they even arrive after ORIF?
- I am not sure what is meant by incidence per 1000 person years etc. I think this concept is more relevant in other types of study designs where we have a more equal follow-up time for all individuals. In this case I think it is much more relevant to focus on cumulative incidence as modelled by a competing risk model. I would thus suggest to remove all references to "incidence" (without cumulative).
- Considering cumulative incidence it seems that death as a competing risk was not considered? I suggest to at least perform some sensitivity analysis with Fine-Gray etc or to motivate why this is not needed.
- I find the statistics method section quite confusing. When was the Fisher’s exact test used for example? Is Stata distributed by SAS institute? Why did you use all those tests in addition to regression? What do you mean by "multivariate" regression? Multivariable/multiple? Or truly multivariate? And did you somehow justify the proportional hazards assumption?
- It is stated that the cumulative rates increased over time. Isn’t this the very definition of cumulative?
- It seems that stratification was based on age, although age was still adjusted for in each strata. Why not use a better model instead? Splines? Interaction effects?
- Some numbers and dates in fig 1 seems to differ from the text.
- Both figures seems to have very low resolution
- Fig 2: Seems to be too long follow-up. I suggest to truncate when the numbers at risk gets too low. I also suggest to add confidence bands or at least pointwise intervals. It is also unclear what is meant by "cumulative incidence". There is no description of a competing risk model so I suppose this is just a reversed Kaplan-Meier?
- Codes in the supplementary table are listed without justification. Are those based on for example Charlson comorbidities? If so, please cite the relevant source. I would also sugest to specify the meaning of "X" in the codes.
- I would like to know more about "Informed consent was obtained from all subjects involved in the study". With over 2 million patients this must have been a huge administrative burden.
Author Response
Dear Editor,
It is really honored of me to resubmit the paper entitled “Long-term Incidence of Total Knee Arthroplasty After Open Reduction and Internal Fixation of Proximal Tibial and Distal Femoral Fractures: A Nationwide Cohort Study” for consideration of publication in the famous journal of yours. Thanks for the reviewer’s reminding, suggestions, and precious opinions to make the article more scientific and complete. We have corrected our mistakes and added detailed description of statistical analysis based on reviewers’ suggestions. We modified our tables and figures with only 1:1 age sex propensity score matching results. This change will more clarify the results of our data and reveal our thinking process for this study by preserving the data into supplementary materials. We found that the patients aged 20–65 years and males had a significantly higher incidence of TKA after ORIF than that in the non-fracture group. The incidence of TKA for the 20–65-year subgroup of the ORIF group was 4% and that for the male subgroup was 1.5%, and the both rates increased over the 17-year follow-up period. We believe that this paper will increase the citation of your precious journal.
All of the authors have read and approved the final revised manuscript. Thank you for your time and effort in reviewing this manuscript. I am looking forward to your reply in your earliest convenience.
With my best wishes!
Comments and Suggestions for Authors
This is a study of patients with ORIF and a possible later TKA. Patients with ORIF are matched with patients without ORIF from the general population in order to study differences in the cumulative incidence rate of TKA. I have a few comments:
- I do not see the need for the first attempt with exact matching based on age sex, and year of surgery. It is concluded that this sample was unbalanced considering comorbidity wherefore propensity scores were used instead. The results are similar, however, wherefore I think the manuscript would be clearer without the initial attempt.
Ans: Thank you for your suggestion. We have renewed our table 1 and figure 2 as the results of propensity match score method. The results of exact matching based on age sex, and year of surgery have been placed us table S2 and figure S1.
- The propensity score procedure is poorly described. Which method was used (logistic regression, GBM etc.)? Greedy matching? Caliper? And especially, what variables were included?
Ans: Thank you for your reminding. We have added the description as below:” Propensity score matching was applied to reduce selection bias between the study groups. Age, sex, index year of surgery, and comorbidities were selected as independent variables. The greedy method was used for matching at a 1:1 ratio between the study groups with a caliper width 0.2-fold the standard deviation of the propensity score between the study groups.
- Who is included in NHIRD? The whole population or just patients who received care during the studied period? If so, at what level? This is rather important for the interpretation of the study!
Ans: Thank you for your reminding. We have added the description as below: “The NHIRD covers 99.7% of the population (nearly 23 million people) in Taiwan. Longitudinal Health Insurance Databases (LHID2000) which randomly sampled 2 million beneficiaries from the original NHIRD in the year of 2000 was adopted for this study. The representativeness of LHIDs has been validated by NHRI.”
- What was the timing of the comorbidity listed in table 1? Were they all present simultaneously as the ORIF or is this based on some look-back period using some records linkage? Or could they even arrive after ORIF?
Ans: Thank you for your reminding. We have added the description as below: “The case definition of each comorbidity was based on diagnostic codes which have been employed and validated in other claims data-based studies. It was determined by the presence of either at least two times of diagnostic codes in outpatient records or one time of discharge codes in hospitalization records one year before the index date.”
- I am not sure what is meant by incidence per 1000 person years etc. I think this concept is more relevant in other types of study designs where we have a more equal follow-up time for all individuals. In this case I think it is much more relevant to focus on cumulative incidence as modelled by a competing risk model. I would thus suggest removing all references to "incidence" (without cumulative).
Ans: Thank you for your suggestion. “Incidence per 1000 person years” means the incidence rate of TKA of one thousand people in one year. We have removed removing all the references to "incidence".
- Considering cumulative incidence, it seems that death as a competing risk was not considered? I suggest to at least perform some sensitivity analysis with Fine-Gray etc. or to motivate why this is not needed.
Ans: Thank you for your suggestion. In our current analysis, we treated death as censored data. This was common in analyzing survival data. In addition, there was no significant difference of death rate between two groups.
- I find the statistics method section quite confusing. When was the Fisher’s exact test used for example? Is Stata distributed by SAS institute? Why did you use all those tests in addition to regression? What do you mean by "multivariate" regression? Multivariable/multiple? Or truly multivariate? And did you somehow justify the proportional hazards assumption?
Ans: We apologized for our wrong descriptions. We have modified the paragraph into: “Continuous variables are presented as means and standard deviations, and categorical variables are presented as number of cases and percentages. Continuous between-group variables were compared using Student’s t-test, and categorical variables were assessed with a chi-square test. These tests were used to compare the characteristics of both groups. Data were evaluated using a log-rank test and univariable and multivariable Cox regression analyses. All statistical analyses were performed using SAS (version 9.4) and Stata 16.1 (StataCorp, College Station, Texas, USA). We considered p values of < 0.05 as statistically significant.” Besides, we have verified that the proportional hazards assumption holds.
- It is stated that the cumulative rates increased over time. Isn’t this the very definition of cumulative?
Ans: Thanks for your reminding. We have removed all references to "incidence" according to your suggestion. We have modified this into “the incidence rates increased over time.
- It seems that stratification was based on age, although age was still adjusted for in each stratum. Why not use a better model instead? Splines? Interaction effects?
Ans: Thanks for your asking. In our analysis we have found the P for interaction of age and sex were significant. Thus, in clinical implication we would like to discuss the impact of fracture receiving surgery on the incidence of TKA of the separate subgroups, respectively.
- Some numbers and dates in fig 1 seem to differ from the text.
Ans: We apologized for our mistakes. We have corrected our dates and numbers in the text as right as the data shown in figure 1.
- Both figures seem to have very low resolution
Ans: We have modified the resolution of the figures.
- Fig 2: Seems to be too long follow-up. I suggest truncating when the number at risk gets too low. I also suggest adding confidence bands or at least pointwise intervals. It is also unclear what is meant by "cumulative incidence". There is no description of a competing risk model, so I suppose this is just a reversed Kaplan-Meier?
Ans: Thanks for your suggestion. In clinical implication we would like to introduce the incident rate during the 19-year follow-up as an important information given by our study. In addition, this is a reversed Kaplan-Meier method.
- Codes in the supplementary table are listed without justification. Are those based on for example Charlson comorbidities? If so, please cite the relevant source. I would also suggest specifying the meaning of "X" in the codes.
Ans: Thanks for your suggestions. The comorbidity-related codes in the supplementary table were based on Charlson comorbidities. We have cited the reference. In addition, we are sorry for our bad descriptions. We have changed the X to the detailed numbers in the codes in table S1.
- I would like to know more about "Informed consent was obtained from all subjects involved in the study". With over 2 million patients this must have been a huge administrative burden.
Ans: We apologize for our wrong description. We have modified in informed consent section as below: “Patient consent was waived due to the National Health Insurance dataset consists of de-identified secondary data used for research purposes, and the Research Ethics Committee of Hualien Tzu Chi Hospital gave a formal written waiver of the need for consent.”
Reviewer 2 Report
There were 4 major revisions and 1 minor revision suggested as below:
Please describe the tracking database (LGTD 2010 for 2 million sampling databases) in Taiwan; explain why could the cases in the present study be followed until Dec. 31, 2018? (According to the statement of LGTD 2010, the sampling files from 2000 were followed up for 18 years (2000 t0 2017))
Please explain what variables did the propensity score match by? (Were all the comorbidities in Table-1 included? Were age and gender still included or not?)
Please explain the reasons to select the comorbidities? Were those comorbidities related to the subsequent TKA?
The present study was a hybrid study (retrospective cohort study) and a case-control study design used. Such cohort-based case-control studies can be divided into two types on the basis of the approach used for selecting the controls. For this present study, it must be suitable to choose the nested case-control study design instead of the case-cohort study design. In this way (a hypothetical nested case-control study), the cases and controls are, in effect, matched on calendar time and length of follow-up (it meant that in Table-2, the person-years would be equal or two times for both groups).
InFigure-2, the K-M methods were plotted in general population and in the male population. Based on the strategical analysis, the K-M plots of the female population could be added.
Author Response
Dear Editor,
It is really honored of me to resubmit the paper entitled “Long-term Incidence of Total Knee Arthroplasty After Open Reduction and Internal Fixation of Proximal Tibial and Distal Femoral Fractures: A Nationwide Cohort Study” for consideration of publication in the famous journal of yours. Thanks for the reviewer’s reminding, suggestions, and precious opinions to make the article more scientific and complete. We have corrected our mistakes and added detailed description of statistical analysis based on reviewers’ suggestions. We modified our tables and figures with only 1:1 age sex propensity score matching results. This change will more clarify the results of our data and reveal our thinking process for this study by preserving the data into supplementary materials. We found that the patients aged 20–65 years and males had a significantly higher incidence of TKA after ORIF than that in the non-fracture group. The incidence of TKA for the 20–65-year subgroup of the ORIF group was 4% and that for the male subgroup was 1.5%, and the both rates increased over the 17-year follow-up period. We believe that this paper will increase the citation of your precious journal.
All of the authors have read and approved the final revised manuscript. Thank you for your time and effort in reviewing this manuscript. I am looking forward to your reply in your earliest convenience.
With my best wishes!
Comments and Suggestions for Authors
There were 4 major revisions and 1 minor revision suggested as below:
Please describe the tracking database (LGTD 2010 for 2 million sampling databases) in Taiwan; explain why could the cases in the present study be followed until Dec. 31, 2018? (According to the statement of LGTD 2010, the sampling files from 2000 were followed up for 18 years (2000 to 2017))
Ans: We apologize for our mistake. The cases in the present study were followed until Dec. 31, 2017. The follow-up period was 17 years. We have corrected this error in the text.
Please explain what variables did the propensity score match by. (Were all the comorbidities in Table-1 included? Were age and gender still included or not?)
Ans: Thank you for your reminding. We have added the description as below:” Propensity score matching was applied to reduce selection bias between the study groups. Age, sex, index year of surgery, and comorbidities were selected as independent variables. The greedy method was used for matching at a 1:1 ratio between the study groups with a caliper width 0.2-fold the standard deviation of the propensity score between the study groups.
Please explain the reasons to select the comorbidities. Were those comorbidities related to the subsequent TKA?
Ans: Thanks for your suggestions. The comorbidity-related codes in the supplementary table were based on Charlson comorbidities. They all may have great impact on the healing of the fracture and the development of post-traumatic arthritis, and the further progression to the indication of TKA surgery.
The present study was a hybrid study (retrospective cohort study) and a case-control study design used. Such cohort-based case-control studies can be divided into two types on the basis of the approach used for selecting the controls. For this present study, it must be suitable to choose the nested case-control study design instead of the case-cohort study design. In this way (a hypothetical nested case-control study), the cases and controls are, in effect, matched on calendar time and length of follow-up (it meant that in Table-2, the person-years would be equal or two times for both groups).
Ans: Thanks for your reminding. This study is a retrospective cohort study but not hybrid study. We have changed the name of the control group as the non-fracture group.
In Figure-2, the K-M methods were plotted in general population and in the male population. Based on the strategical analysis, the K-M plots of the female population could be added.
Ans: Thanks for your suggestion. The incidence rates had no significant difference in the females. So, we did not add the K-M plots of the female population.

Round 2
Reviewer 1 Report
The manuscript has improved but some questions remains:
- The use of LHID2000 has been clarified. It is stated online, however (https://nhird.nhri.org.tw/en/Data_Subsets.html), that this dataset includes 1 million patients. I would sugest to include a reference clarifying the numbers. Also, even if 2 million patients are included, it is also stated that 2,000,124 patients were included in the study. Were some patients included more than once (bilateral cases)?
- I still suggest to remove everything related to the exact matching (not only in the result table). This includes abstract, fig 1, methods section etc. It could be kept in a supplement of course (as referenced from the discussion section), but I still don’t think it adds much value.
- The statistical section should mention that Kaplan-Meier was used for survival analysis (the log-rank test is specified but also the method for the illustrated curves needs some clarification in the text).
- The mean follow up time was previously mentioned in the text but has now been removed. I agree that this number is not an adequate number to present, but I would prefer to have a more relevant number instead. For example the median of a reversed Kaplan-Maier (method 5 according to https://cemsiis.meduniwien.ac.at/fileadmin/user_upload/_imported/fileadmin/msi_akim/CeMSIIS/KB/volltexte/Schemper_Smith_1996_CCT.pdf).
- I still find references to incidence confusing. Note that I did not ask to remove "cumulative" but rather to avoid the use of "incidence" without a specification of "cumulative". I did this since I interpreted the analysis as involving a competing risk analysis. It was stated in the rebuttal that the included survival analysis is based solely on Kaplan-Meier, however. I therefore think that terminology associated with such analysis should be used throughout instead. This aplies to all parts of the text, as well as figure captions and axis labels. I am not able to interpret "incidence" (or the rate per person year as stated on line 121) when those person years are distributed unequally among patients. Thus, please reformulate the outcome in terms of probabilities within certain time points.
- The estimate at 17 years seems extremely unreliable. If this is a relevant measure (as stated in the rebuttal), more patients needs to be included with this follow-up time. Otherwise the last part of fig 2 simply makes no sense, at least not without corresponding measures of uncertainty (confidence bands or point wise CIs) and with a corresponding table indicating patients at risk. It also makes no sense to refer to those numbers without CIs in the text. It seems to me that very few (if any) patients were followed for at least 17 years (even though the inclusion period was 17 years, which is not the same as the length of follow up for individual patients).
SMALLER ISSUES:
- The description of the propensity score method has improved. However, the over-all method (i.e. logistic regressin, GBM, XGBoost etc) is still not specified.
- In the last version SAS institute was specified as the software vendor for Stata. This was now changed but the vendor for SAS is now missing instead.
- All point estimates should have CIs as well. This is missing for example on line 125.
- The term "multivariate" was changed to multivariable in the text but not in table 2. Also if all models are univariate, the term "univariate" should probably be changed to "simple" or "univariable" in accordance.
- The control group has been renamed in the text but not in the tables.
- There are p-values for interaction in table 3 but it is not specified what those refers to.
- Line 126 seems to fit better in the methods section.
Author Response
Dear reviewers:
Thanks for your suggestions to make our article more reliable and scientific. We have addressed all your comments carefully and give out our responses and modification to our manuscript. About your Major Q6 problem we would add confidence bands and number at risk in Figure 2. It would be similar to the example shown below. However, it would take some time (about two weeks) to modify the figure since we need to apply to enter the HWDC to perform analysis and the results needed to be approved by the staff of HWDC. Thanks for your patience.
Sincerely yours,
The manuscript has improved but some questions remains:
- The use of LHID2000 has been clarified. It is stated online, however (https://nhird.nhri.org.tw/en/Data_Subsets.html), that this dataset includes 1 million patients. I would suggest including a reference clarifying the numbers. Also, even if 2 million patients are included, it is also stated that 2,000,124 patients were included in the study. Were some patients included more than once (bilateral cases)?
Ans: Thanks for your kind reminding. Initially, National Health Insurance Research Database (NHIRD) was maintained by National Health Research Institute (NHRI). The website you cited did claim that LHID200 consisted of 1 million patients. However, it was transferred to the Health and Welfare Data Science Center (HWDC) of the Ministry of Health and Welfare from 2016/06/28. The government decided extending the number of subjects to 2 million at meanwhile. According to description from the HWDC (https://dep.mohw.gov.tw/dos/cp-2506-3633-113.html), the LHID2000 contains 2,000,118 people. We had modified the number in Figure 1.
PS We are sorry that the website is in Chinese. However, the government provides detail description of LHID2000 in English (Word file). We could provide it if needed.
- I still suggest removing everything related to the exact matching (not only in the result table). This includes abstract, fig 1, methods section etc. It could be kept in a supplement of course (as referenced from the discussion section), but I still don’t think it adds much value.
Ans: Thanks for your suggestion. We have removed all the related descriptions from Method, Results, and figure 1 and added them to the discussion section.
- The statistical section should mention that Kaplan-Meier was used for survival analysis (the log-rank test is specified but also the method for the illustrated curves needs some clarification in the text).
Ans: Thanks for your kind reminding. We had specified the methods in detail as below: “Survival curves were estimated according to the Kaplan–Meier procedure, and groups were compared with use of the log-rank test.”
- The mean follow-up time was previously mentioned in the text but has now been removed. I agree that this number is not an adequate number to present, but I would prefer to have a more relevant number For example, the median of a reversed Kaplan-Maier (method 5 according to https://cemsiis.meduniwien.ac.at/fileadmin/user_upload/_imported/fileadmin/msi_akim/CeMSIIS/KB/volltexte/Schemper_Smith_1996_CCT.pdf).
Ans: Thanks for your suggestion. However, the proportion of subjects receiving TKA in our study was very low (1~4%) for both groups even after long follow-up time. It would be difficult to get the median of a reversed Kaplan-Meier. Thus, we decided to remove the follow-up time to avoid confusing the readers. We hope that the reviewer could agree with that.
- I still find references to incidence confusing. Note that I did not ask to remove "cumulative" but rather to avoid the use of "incidence" without a specification of "cumulative". I did this since I interpreted the analysis as involving a competing risk analysis. It was stated in the rebuttal that the included survival analysis is based solely on Kaplan-Meier, however. I therefore think that terminology associated with such analysis should be used throughout instead. This applies to all parts of the text, as well as figure captions and axis labels. I am not able to interpret "incidence" (or the rate per person year as stated on line 121) when those person years are distributed unequally among patients. Thus, please reformulate the outcome in terms of probabilities within certain time points.
Ans: We agree with your suggestion. We had modified the label of y axis in KM curve into “Proportion of subjects receiving TKA” according to the reference below. In addition, we also modified the term “incidence” in the text into “proportion of subjects receiving TKA” in all parts of the text.
Reference:
- Heart Outcomes Prevention Evaluation Study Investigators, Yusuf S, Sleight P, Pogue J, Bosch J, Davies R, Dagenais G. Effects of an angiotensin-converting-enzyme inhibitor, ramipril, on cardiovascular events in high-risk patients. N Engl J Med. 2000 Jan 20;342(3):145-53. doi: 10.1056/NEJM200001203420301. Erratum in: 2000 May 4;342(18):1376. Erratum in: N Engl J Med 2000 Mar 9;342(10):748. PMID: 10639539.
- The estimate at 17 years seems extremely unreliable. If this is a relevant measure (as stated in the rebuttal), more patients need to be included with this follow-up time. Otherwise, the last part of fig 2 simply makes no sense, at least not without corresponding measures of uncertainty (confidence bands or point wise CIs) and with a corresponding table indicating patients at risk. It also makes no sense to refer to those numbers without CIs in the text. It seems to me that very few (if any) patients were followed for at least 17 years (even though the inclusion period was 17 years, which is not the same as the length of follow up for individual patients).
Ans: Thanks for your kind reminding. We agree with your suggestion. We would add confidence bands and number at risk in Figure 2. It would be similar to the example shown below. However, it would take some time (about two weeks) to modify the figure since we need to apply to enter the HWDC to perform analysis and the results needed to be approved by the staff of HWDC.
SMALLER ISSUES:
- The description of the propensity score method has improved. However, the over-all method (i.e., logistic regression, GBM, XGBoost etc.) is still not specified.
Ans: We have described our method as below: “Propensity scores matching method was applied to reduce selection bias between the study groups. Age, sex, index year date of surgery, and comorbidities enrolled in CCI calculated at a 1:1 ratio. were selected as independent variables. The greedy method was used for matching at a 1:1 ratio between the study groups with a caliper width 0.2-fold the standard deviation of the propensity score between the study groups. The study outcome was subsequent TKA performed after index surgery for distal femoral or proximal tibial factures. All available data were included in this study, and no additional unpublished data were included. The subgroup analysis was then performed by dividing all the patients into two groups by age (20–65 years and > 65 years) and by sex (male and female), respectively. Interaction tests were employed to determine the subgroup effects of age and sex on the TKA risk.
Continuous variables are presented as means and standard deviations, and categorical variables are presented as number of cases and percentages. Continuous be-tween-group variables were compared using Student’s t-test, and categorical variables were assessed with a chi-square test. These tests were used to compare the characteristics of both groups. Data were evaluated using univariable and multivariable Cox regression analyses. Survival curves were estimated according to the Kaplan–Meier procedure, and groups were compared with use of the log-rank test. All statistical analyses were performed using SAS version 9.4 (SAS Institute Inc., Cary, NC, USA) and Stata 16.1 (StataCorp, College Station, Texas, USA). We considered p values of < 0.05 as statistically significant.”
- In the last version SAS institute was specified as the software vendor for Stata. This was now changed but the vendor for SAS is now missing instead.
Ans: Sorry for our mistakes. We have now marked it as below:” SAS version 9.4 (SAS Institute Inc., Cary, NC, USA)”
- All point estimates should have CIs as well. This is missing for example on line 125.
Ans: Thanks for your reminding. We have modified our sentence as below: “From the Cox proportional hazard regression model with adjustments for all baseline characteristics shown in Table 1, we observed that the adjusted hazard ratio (aHR) de-rived for the ORIF group relative to the non-fracture group was 1.23 (1.07–1.41; table 2).”
- The term "multivariate" was changed to multivariable in the text but not in table 2. Also if all models are univariate, the term "univariate" should probably be changed to "simple" or "univariable" in accordance.
Ans: Sorry for our mistakes. We have modified all the terms in our text and table into multivariable and univariable.
- The control group has been renamed in the text but not in the tables.
Ans: Sorry for our mistakes. We have modified all the terms in our text and table into non-fracture group instead of control group.
- There are p-values for interaction in table 3 but it is not specified what those refers to.
Ans: Thanks for your reminding. We have specified the meaning of p for interaction in the method section as below: “…Interaction tests were employed to determine the subgroup effects of age and sex on the TKA risk.” And the Result section as below: “The effects of age and sex on the TKA risk were significant based on p for interaction results (table 3).”
- Line 126 seems to fit better in the methods section.
Ans: Thanks for your reminding. We have modified it into the method section.
Reviewer 2 Report
The present study was a retrospective cohort study and a case-control study design used. Such cohort-based case-control studies can be divided into two types on the basis of the approach used for selecting the controls. For this present study, it must be suitable to choose the nested case-control study design instead of the case-cohort study design. In this way (a hypothetical nested case-control study), the cases and controls are, in effect, matched on calendar time and length of follow-up (it meant that in Table-2, the person-years would be equal for both groups). On the other hand, there was a great difference of 25,389 person-years but only a small difference of 38 TKA cases noted between both groups. It should be not reasonable.
Therefore, the non-fracture group must be selected by propensity score matched by age, gender, index month of surgery and other comorbidities. The most important variable was index month of surgery (in the same year), not only index year of surgery.
Figure-1 should be revised to show the 1:1 matched non-fracture group only since another control group (1:2 matched non-fracture group) was not analyzed subsequently; the match method was also revised as “matched by propensity score based on age, gender, index month of surgery, comorbidities enrolled in CCI calculated”.
What was the correct number of case group (fracture group)? 32627 in Figure-1 or 32592 in Table-1?
Because of the above commendations, all the data of control group (1:1 matched non-fracture group) must be re-calculated and re-analyzed. Focused on Table-2 again, it was not reasonable that there was a great difference of 25,389 person-years but only a small difference of 38 TKA cases noted between both groups. All the subsequent results would be not reliable.
Author Response
Thanks again for the reviewer’s reminding and suggestions for making our work more improved and correct. We have addressed every comment carefully and corrected our mistakes. Hope that we have met your criteria for acceptance in your precious journal.
All the authors have agreed the changes in our revised manuscript and declared no conflict of interest.
Sincerely yours,
Response to Reviewer 2:
The present study was a retrospective cohort study and a case-control study design used. Such cohort-based case-control studies can be divided into two types on the basis of the approach used for selecting the controls. For this present study, it must be suitable to choose the nested case-control study design
instead of the case-cohort study design. In this way (a hypothetical nested case-control study), the cases and controls are, in effect, matched on calendar time and length of follow-up (it meant that in Table-2, the person-years would be equal for both groups). On the other hand, there was a great difference of 25,389 person-years but only a small difference of 38 TKA cases noted between both groups. It should be not reasonable. Therefore, the non-fracture group must be selected by propensity score matched by age, gender, index month of surgery and other comorbidities. The most important variable was index month of surgery (in the same year), not only index year of surgery.
Ans:
- Thanks for reviewer’s suggestion. However, our study was a retrospective cohort study as shown below. We enrolled patients with incident distal femur and proximal tibia fractures as exposed group, whereas patients without distal femur and proximal tibia fractures as unexposed group. Taking advantage of 17-year follow-up during 2001–2017, we could compare the difference of long-term risk of TKA incidence between two groups. According to the Cox regression, we could observe that exposed group had significant higher hazard ratio (HR=1.23) than unexposed group. Due to the large sample size and long follow-up period, it could result in great difference of person years between two groups. In addition, we had double confirmed the correctness of data.
- Sorry for our mistake. The non-fracture group was selected by propensity score matched by age, gender, index date of surgery and other comorbidities.
Figure-1 should be revised to show the 1:1 matched non-fracture group only since another control group (1:2 matched non-fracture group) was not analyzed subsequently; the match method was also revised as “matched by propensity score based on age, gender, index month of surgery, comorbidities enrolled in CCI calculated”.
Ans: Thanks for reviewer’s reminding. We have modified the Figure 1 based on your suggestion and modified the description of math method as “matched by propensity score based on age, gender, index date of surgery, comorbidities enrolled in CCI calculated” in the manuscript.
What was the correct number of case group (fracture group)? 32627 in Figure-1 or 32592 in Table-1?
Ans: Sorry for our mistake. The correct number of case group was 32592. We have modified the Figure-1.
Because of the above commendations, all the data of control group (1:1 matched non-fracture group) must be re-calculated and re-analyzed. Focused on Table-2 again, it was not reasonable that there was a great difference of 25,389 person-years but only a small difference of 38 TKA cases noted between both groups. All the subsequent results would be not reliable.
Ans: Thanks again for reviewer’s suggestion. We had specified that our study design was a nation-wide based cohort study. We adopted this study design so that we could take advantage of long-term follow-up and compared the difference of long-term risk of TKA incidence between two groups. Due to the large sample size and long follow-up period, it could result in great difference of person years between two groups.